# Improving the Mechanical Properties of CCFRPLA by Enhancing the Interface Binding Energy and Strengthening the Anti-Separation Ability of a PLA Matrix

**DOI:** 10.3390/polym15112554

**Published:** 2023-06-01

**Authors:** Hongbin Li, Zhihua Li, Na Wang, Yubao Peng, Zhuang Jiang, Qiushuang Zhang

**Affiliations:** 1Department of Electromechanical Engineering, Qingdao University of Science and Technology, Qingdao 266061, China; qklhb1989@qust.edu.cn (H.L.); lizhihua127@163.com (Z.L.); m17861435229@163.com (Y.P.); jiangz335553@163.com (Z.J.); 2School of Mechanical Engineering, Tianjin University, Tianjin 300354, China

**Keywords:** continuous fiber-reinforced polymers, ultrasonic vibration, mechanical properties, interface binding energy, molecular dynamics

## Abstract

Additive manufacturing (AM) can produce almost any product shape through layered stacking. The usability of continuous fiber-reinforced polymers (CFRP) fabricated by AM, however, is restricted owing to the limitations of no reinforcing fibers in the lay-up direction and weak interface bonding between the fibers and matrix. This study presents molecular dynamics in conjunction with experiments to explore how ultrasonic vibration enhances the performance of continuous carbon fiber-reinforced polylactic acid (CCFRPLA). Ultrasonic vibration improves the mobility of PLA matrix molecular chains by causing alternative fractures of chains, promoting crosslinking infiltration among polymer chains, and facilitating interactions between carbon fibers and the matrix. The increase in entanglement density and conformational changes enhanced the density of the PLA matrix and strengthened its anti-separation ability. In addition, ultrasonic vibration shortens the distance between the molecules of the fiber and matrix, improving the van der Waals force and thus promoting the interface binding energy between them, which ultimately achieves an overall improvement in the performance of CCFRPLA. The bending strength and interlaminar shear strength of the specimen treated with 20 W ultrasonic vibration reached 111.5 MPa and 10.16 MPa, respectively, 33.11% and 21.5% higher than those of the untreated specimen, consistent with the molecular dynamics simulations, and confirmed the effectiveness of ultrasonic vibration in improving the flexural and interlaminar properties of the CCFRPLA.

## 1. Introduction

Continuous fiber-reinforced polymers (CFRP) have been widely used in aerospace, rail transit, and marine equipment because of their excellent features, such as high specific strength and modulus, preferable compressive stability, and good design performance [1,2,3]. Additive manufacturing (AM) could fabricate complex configuration structures that were difficult to fabricate via conventional machining methods through layered stacking. CFRP produced by AM technology achieves a decoupling between product complexity and manufacturing cost, promoting the widespread use of structurally functional integrated CFRP in unmanned aerial vehicles and satellites [3,4,5,6]. As one of the most promising bio-degradable polymers, polylactic acid (PLA) has been widely used in the fused filament fabrication (FFF) process as feedstock or matrix to fabricate CFRP due to its excellent environmental friendliness and mechanical properties [7,8,9], and the PLA/CFRP composites could be printed by a dual extrusion or co-extrusion method based on AM [10,11]. Moreover, given the excellent physical properties of CCFRPLA, it is widely used in aerospace, automotive, wind energy, and sports fields [9,10,11]. However, owing to the limitations of no reinforcing fibers in the lay-up direction and weak interface bonding between fibers and the matrix, CCFRPLA fabricated by AM technology can only perform in a certain direction in most cases, and their range of use as key load-bearing components is limited [12,13,14]. The surfaces of continuous fibers, especially carbon fibers, are inert and lack functional groups that interact with the polymer matrix, resulting in weak interfacial bonding between the two through intermolecular forces [3]. The interface between the fiber and matrix serves as a bridge for load transfer, which is a vulnerable point to failure under external loads, and its performance significantly affects the overall properties of the CFRP [15,16]. Methods such as fiber surface modification and chemical grafting can generate active functional groups on the fiber surface that can easily interact with the matrix [3,17,18], which improves the roughness of the fiber surface and changes the connection between the fiber and matrix, thereby improving the interface performance. However, these methods damage not only the integrity of the fiber, reducing its load-bearing capacity, but also the residual chemical reagents on the surface of the fiber, causing damage to the matrix polymer. Laser heating, tempering, and hot pressing can improve the interfacial performance of the matrix by increasing the time for which the matrix is above the glass-transition temperature [19,20], promoting the degree of infiltration, and extending the interaction time between the molecular chains. By increasing the recrystallization degree of the interface between the fibers and matrix, the interfacial performance can be improved, ultimately improving the overall performance of the CFRP. Andreu added a heated roller to the printer to increase interlayer adhesion [21]. In this process, the added heated roller could apply homogeneous pressure forces to increase filament surface contact, and the applied heat could enable longer diffusion and neck growth between adjacent filaments. Todoroki adopted a stepwise manufacturing approach to overcome the defect of no reinforcing fibers in the lay-up direction [22]. In this method, a composite part with a through hole and a reinforcing bar with continuous carbon fibers were printed separately. Subsequently, the reinforcing bar was inserted into the through hole, and the two parts were fused together via resistive heating. The topology optimization of the structure and fiber distribution is an effective method for enhancing the overall performance of CFRP [23,24]. Topology optimization of the product structure and printing process can be performed according to the stress distribution of the composites under actual working conditions, and the fiber distribution inside the product can be optimized according to the stress of the product, which can improve the overall bearing capacity of the product. Ultrasonic vibration during FFF printing was a non-chemical/non-thermal process, and there were no adverse effects on the final product [11]. Chen developed a printer head that vibrated the nozzle vertically to improve the vertical tensile strength of FFF parts [25]. The vibration in the vertical direction reduced the viscosity of the material and brought downward inertial force, enhancing the diffusion of the polymer chains and strengthening the inter-layer bonding strength and mechanical properties of FFF parts. Tofangchi explored the effect of ultrasonic vibration on the interlayer adhesion of acrylonitrile butadiene styrene (ABS) and found that ultrasonic vibrations could result in an increase of up to 10% in the interlayer adhesion. This increase was attributed to the increase in polymer reptation due to ultrasonic vibration-induced relaxation of the polymer chains from secondary interactions in the interface regions [26]. Maidin et al. [27] supplied the vibration onto the printing platform to identify the impact of ultrasound vibration on the ABS polymer, and the experimental results indicated that both the tensile strength and Young’s modulus improved by 11.3% and 16.7% compared to the untreated specimen [28,29]. Previous studies have found that ultrasonic vibration can improve the activity of the matrix molecular chains, shorten the creep-relaxation time, promote the infiltration and cross-linking of polymer molecular chains, and improve the interlayer bonding performance of the matrix in CFRP [30]. Based on previous research, this study explores the influence mechanism of ultrasonic vibration on the interface properties between fibers and the matrix and studies the influence of ultrasonic vibration on the overall mechanical properties of CFRP produced based on AM. This study lays a theoretical and experimental foundation for research on ultrasonic online vibration-assisted additive manufacturing of CFRP.

## 2. Experimental

### 2.1. Sample Preparation

In this study, an ultrasonic online vibration-assisted AM platform was designed to fabricate CCFRPLA composites, as shown in Figure 1a. In the designed ultrasonic online vibration-assisted AM platform, the ultrasonic transducer was connected to the heating block through a specially designed polyether-ether-ketone connector to prevent damage to the ultrasonic transducer caused by the high temperature of the heating block. There was a through-hole in the center of the heating block through which the continuous fibers passed during printing. A slanted hole on the side at 45° from the center hole was designed, and the polymer matrix entered through this hole and melted in the mixing chamber during printing, as shown in Figure 1b. In the co-extrusion process of CCFRPLA, the PLA resin filament and the continuous carbon fibers were separately supplied to the printer head. The PLA filament became molten inside the heated nozzle, and when the continuous carbon fibers were passed through the nozzle, it got impregnated by the PLA resin and extruded from the bottom nozzle, as shown in Figure 1c. Under the action of ultrasonic vibration, the melt polymer could sufficiently soak with continuous fibers, improving the infiltration effect between fibers and PLA matrix.

The fixed frequency of the ultrasonic transducer used in this study was 30 KHz, and the power was adjusted to within 30 W. The PLA (Shenzhen Creativity 3D Technology Co. Ltd., Shenzhen, China) was chosen as the resin matrix material in this study owing to its excellent mechanical properties and environmental friendliness [31,32], and carbon fiber (6K) (SINOSTEEL JILIN CARBON CO., LTD., Shenzhen, China) with a tensile strength of approximately 3530 MPa was utilized as the reinforcement. Experimental CFRP specimens were fabricated under ultrasonic powers of 0, 10, 20, and 30 W using the designed ultrasonic vibration-assisted AM platform with the parameters listed in Table 1.

### 2.2. Experimental Procedure and Results

Tensile and interlaminar shear tests were carried out in accordance with ISO 527:1997 and ISO 14130:1997, respectively, where the CCFRPLA specimen dimensions for the tensile test were 250 mm × 14 mm × 2 mm, and the fiber direction in the specimen was the same as the tensile direction. The specimen dimensions for the interlaminar shear test were 20 mm × 10 mm × 2 mm, and the direction of the carbon fiber in the CFRP was consistent with the long side of the specimen. In addition, the bending strength of specimens under different ultrasonic powers was studied, and the specimen for the three-point bending test was of a non-standard design with the dimensions 60 mm × 10 mm × 2 mm. The fibers were printed in the direction of 0°, corresponding to the longest side of the specimen. During testing, a vertical force was applied at the longitudinal midpoint at a loading rate of 2 mm/min. The displacements of the indenter and loads applied to the specimen during the experiment were recorded, and the results are shown in Figure 2.

## 3. Discussion

### 3.1. The Effect of Ultrasonic Vibration on Mechanical Properties of CFRP

The experimental results illustrate that the introduction of ultrasonic vibration during the CFRP preparation process can significantly improve the mechanical properties of the composite. As the ultrasonic vibration power increased, the macroscopic tensile and bending properties of the CFRP specimens improved. When the ultrasonic power was 20 W, the tensile strength and bending strength of the specimens reached maximum values of 166.8 MPa and 111.5 MPa, respectively, which were 5.6% and 30.11% higher than those of the untreated part. However, when the ultrasonic power reached 30 W, the performance of the specimen was somewhat lower than the maximum value but still much stronger than that of the untreated specimen. This may be due to excessive ultrasonic power causing continuous fiber breakage, resulting in a decline in performance, and the interfacial shear strength coinciding with the macroscopic properties.

The macroscopic properties of composite materials depend on the interfacial properties between the carbon fiber and the matrix, the conformation of the PLA molecular chains, and the microstructure of the matrix itself. The influence of ultrasonic vibration on the conformation of PLA molecular chains, internal microstructure, and properties, and the resulting impact on the macroscopic properties of PLA/CF composites, were difficult to obtain through the aforementioned macroscopic experiments. Molecular dynamics (MD) can be used to explore the effects of ultrasonic vibration on the conformation and activity of PLA molecular chains at the molecular scale [33,34,35], as well as the resulting effects on the microstructure and properties of the interlamination and carbon fiber/matrix interface, thereby revealing the mechanism of ultrasonic vibration on CFRP.

### 3.2. MD Models and Ultrasonic Vibration Settings

In the ultrasonic vibration-assisted FFF manufacturing platform designed in this study, ultrasonic vibration acted directly on the PLA substrate through the nozzle, and the vibration effect was then transmitted to the carbon fiber through the PLA molecular chains. During the transmission process of ultrasonic vibration, an alternative fracture of the PLA matrix molecular chains may occur, leading to changes in the matrix motion activity, fiber/matrix interaction, and permeability. Therefore, before conducting the MD analysis, it is necessary to first determine the weight of the PLA matrix molecular chain. The effect of ultrasonic vibration on the molecular weight was determined through gel permeation chromatography (GPC), and the results are shown in Figure 3.

As shown in Figure 3, ultrasonic vibration altered the molecular weight (MW) of the PLA matrix. As the ultrasonic power increased, the MW of the matrix gradually decreased, indicating that the PLA molecular chain underwent fracture under the action of ultrasonic vibration, which resulted in different lengths of molecular branches and provided an experimental basis for subsequent MD modeling.

According to the GPC results and considering the effect of ultrasonic vibration on the molecular chains of PLA matrix, three different PLA molecular chains with polymerization degrees of 3 (PLA-3), 6 (PLA-6), and 10 (PLA-10) were constructed in this study, and the corresponding molecular dynamics models with these three molecular chain lengths were constructed in the commercial software Materials Studio, as shown in Figure 4a. In the MD models, graphene was utilized as a substitute for CF. The size of the model is 21.3 × 24.6 × 23.4 Å^3^, α = β = γ = 90°, and the density of PLA is 1.25 g/cm^3^. The parameters of the three MD calculation models were the same, except for the different degrees of polymerization of the PLA molecular chains.

Before conducting MD calculations, a standard layer thickness surface in the (001) direction was cut based on the constructed PLA/CF model, and then a 20 Å vacuum layer was added on the surface as the activity space of the molecular chain after ultrasonic vibration, as shown in Figure 4b. Geometry optimization with 10,000 iteration steps (Δt = 1 fs) was first performed on the prepared model, and then a 10 ps dynamic relaxation was performed on the optimized structure in the COMPASS II force field with constant volume, temperature, and number of particles (NVT) to release the residual stress and obtain low-energy conformations [21]. The equilibrium structure with the lowest energy was obtained as the initial structure for subsequent ultrasonic vibration treatment, as shown in Figure 4b. A certain initial velocity was assigned to the moving part at the surface of the relaxed model in the Z-direction, as exhibited in the yellow part of Figure 4c, representing the effect of ultrasonic vibration. Subsequently, a dynamic calculation of 50 ps with a time step of 1 fs was performed on the model in the NVT ensemble. After the calculation was completed, the last frame of the process was taken, and an initial velocity opposite to the previous direction was applied to the same PLA molecular chains to continue another 50 ps dynamic calculation under the same conditions, as shown in Figure 4d. Changing the initial velocity assigned to the PLA molecular chains to explore the effect of ultrasonic powers on the performance of CCFRPLA, and three different initial velocities of 20, 40, and 60 m/s were set to conduct the above molecular dynamics research.

### 3.3. The Interface Binding Energy between CF and Matrix

The interface of CFRP, as a key part of the composite, is vulnerable to failure under a load, and its structural performance plays a crucial role in the overall performance of the composite. Based on the MD model treated with ultrasonic vibration, the interface performance between the carbon fiber and the PLA matrix was explored at the molecular scale. The interface binding energy is an important indicator for characterizing the bonding strength [33]. To explore the impact of ultrasonic vibration on the interface binding energy between the carbon fiber and the PLA matrix, the fiber in the MD model was extracted from the PLA matrix according to Reference [36], as shown in Figure 5a. Figure 5a shows the change in the interface binding energy during the extraction process. 

The interaction energy between CF and matrix PLA is calculated by Equation (1) [36]:(1)$$\Delta E=E_{total}-\left(E_{CF}+E_{PLA}\right)$$
where $E_{total}$ refers to the total energy of the entire system, $E_{CF}$ refers to the energy of carbon fiber, and $E_{PLA}$ refers to the energy of PLA.

According to Equation (1), the binding energy between the carbon fiber and matrix during the extraction process can be determined, and the maximum statistical value is taken as the interface binding energy between the fiber and matrix. In this study, the interfacial binding energy between the PLA matrix and fibers with three different degrees of polymerization under three different speed treatments was calculated (Figure 5b).

As shown in Figure 5b, ultrasonic vibration improved the interfacial binding energy of the composites to a certain extent, and the improvement increased with the ultrasonic vibration amplitude. When the initial velocity of the simulated ultrasonic effect was 60 m/s, the interface binding energies of PLA-3, PLA-6, and PLA-10 with CF increased by 8.0%, 8.42%, and 10.49%, respectively, compared with the specimens without ultrasonic vibration.

The interfacial bonding force between the fiber and matrix is mainly composed of van der Waals and intermolecular electrostatic forces. Analysis of the change in interfacial binding energy components during the extraction process of CF revealed that the change in interfacial energy caused by the van der Waals force was consistent with the change in interfacial binding energy between the CF and PLA matrix, and the difference between the two was approximately 5 kcal/mol. However, the change in the interface binding energy caused by the electrostatic force was negligible, as shown in Figure 6.

The interface binding energy between the CF and PLA matrix was mainly contributed by van der Waals forces. The van der Waals force between the CF and PLA matrix is mainly determined by the molecular spacing between the two. Subsequently, the effect of ultrasonic vibration on the distance between the CF and PLA molecules was investigated, and the results are shown in Figure 7.

As shown in Figure 7, ultrasonic vibration had a significant impact on the distance between the CF and the PLA matrix, and the degree of influence was related to the molecular weight and ultrasonic strength. As the ultrasound amplitude increased, the distance between PLA and CF gradually decreased. At an ultrasonic speed of 60 m/s, the distance between PLA molecules with a degree of polymerization of 3 and CF decreased by 1.18 Å, a decrease of 26.51% compared to untreated PLA molecules. At the same speed, the molecular distance between PLA-6 and CF decreased by 2.57 Å, a decrease of 41.72%, while the molecular distance between PLA molecules with a degree of polymerization of 10 and CF decreased by 3.05 Å, a decrease of 42.72%. At an ultrasonic speed of 20 m/s, the interface spacings between PLA-3, PLA-6, and PLA-10 and CF decreased by 0.63 Å, 1.09 Å, and 0.98 Å, respectively. Compared to the situation at 60 m/s, the decrease in the interface distance was relatively small. The decrease in the intermolecular distance between the CF and PLA matrix reacted directly with an increase in the van der Waals force between the CF and PLA, which further increased the interface binding energy between them. In addition, it was not difficult to see from Figure 7 that PLA molecules with different degrees of polymerization had different sensitivities to ultrasonic vibration. As the degree of polymerization of the PLA molecular chains increased, the effect of ultrasonic vibration on the interface distance became increasingly apparent. This was mainly because, as the degree of polymerization of the PLA molecular chains increased, the radius of gyration of the PLA chains increased, resulting in an increase in the distance between adjacent PLA molecules and the porosity inside the PLA matrix. Ultrasonic vibration can alter the conformation of molecular chains, promote relaxation and fusion between molecular chains, reduce internal pores in the matrix, improve molecular chain activity, and promote the interaction between matrix molecules and CF. However, PLA molecular chains with lower degrees of polymerization are less affected by the molecular chain structure because of their smaller radius of gyration and can form denser polymers, causing a weak impact of ultrasonic vibration on the interface distance. This phenomenon coincides with the interfacial binding energy between PLA and CF at different polymerization degrees and ultrasonic speeds (Figure 5b).

Changes in the conformation and activity of the PLA molecular chains caused by ultrasonic vibration can affect the entanglement infiltration inside the matrix, thereby altering the microstructure inside the matrix material and affecting its ability to withstand loads. According to the change in the PLA matrix energy during CF/PLA extraction, ultrasonic vibration not only improved the interface binding energy between the CF and PLA matrix, but also improved the deformation energy of the PLA matrix (Figure 8).

As shown in Figure 8, ultrasonic vibration also improved the energy of the matrix, which promoted the anti-separation ability of the matrix and contributed to the performance improvement of the CFRP. When the ultrasonic speed was 60 m/s, compared to the model without ultrasonic treatment, the energies of the PLA-3, PLA-6, and PLA-10 matrices increased by 19.8%, 9.4%, and 6.4%, respectively. The matrix energy improvement owing to ultrasonic vibration was mainly attributed to the enhanced activity of the matrix, which changed the conformation of the PLA molecular chains, promoted entanglement and infiltration diffusion between the PLA molecular chains, and finally strengthened the anti-separation ability of the matrix. In addition, the improvement in the matrix energy with different degrees of polymerization varied with the change in ultrasonic speed, which was mainly due to the degree of influence of the ultrasonic vibration on the conformation and density of the PLA molecular chains (Figure 9). 

As shown in Figure 9a, with an increase in the ultrasonic vibration amplitude, the conformation of the PLA molecular chains changes and the PLA molecular chains become more stretched, which promotes entanglement among the molecular chains, reducing the porosity inside the matrix, and increasing the density of the matrix. Figure 9b,c shows the change in the thickness and maximum density of the PLA-3 matrix with the ultrasonic vibration amplitude. As shown in the results, with an increase in the ultrasonic amplitude, the thickness of the PLA-3 matrix decreased and the density accordingly increased, indicating an increase in the degree of crosslinking and density within the matrix. This result coincides with the PLA matrix energy, as shown in Figure 8, revealing the mechanism by which ultrasonic vibration improves the matrix’s performance. In addition, changes in entanglement among the molecular chains and the interaction between CF and the PLA matrix were also related to the activity of the molecular chains. Ultrasonic vibration significantly improved the activity and diffusion coefficient of PLA molecular chains. Figure 10 shows the mean squared displacement (MSD) of PLA molecular chains with different ultrasonic velocities.

Affected by the conformations of the PLA molecular chains with different degrees of polymerization, ultrasonic vibration showed different levels of impact on the activity and diffusivity of the PLA molecular chains. PLA-3, owing to its lower degree of polymerization, entanglement, and entanglement between molecular chains, is more susceptible to ultrasonic vibration, resulting in the greatest MSD values and diffusion coefficients, and the strongest interaction with CF and other molecular chains, which shared similarities with the SEM of the CFRP cross-section (Figure 11).

As Figure 11a–c illustrates, with ultrasonic power increasing, there was more PLA matrix attaching to the fiber surface, indicating the improvement of bonding force between CF and PLA resin with the increase of ultrasonic power, which catered to the binding energy shown in Figure 5. Furthermore, the infiltration between the CF bundle and PLA matrix and the uniformity of CF fiber dispersion in the matrix significantly improved as the ultrasonic power increased, demonstrating the effectiveness of ultrasonic vibration in enhancing the mechanical properties of CFRP, which coincided with the properties of CCFRPLA exhibited in Figure 2. However, when the ultrasonic power increased to 30 W, the resin attached to the fiber surface and the wettability between CF and PLA matrix decreased at the side of 20 W, and there was a fracture inside the fiber bundle, which meant that excessive ultrasonic power could weaken the interface between fibers and matrix, as well as the integrity of CCFRPLA composites, and this result shared the same coincidence with Figure 2.

## 4. Conclusions

This study experimentally explored the effect of ultrasonic vibrations on the performance of 3D-printed CCFRPLA in conjunction with MD simulations. The experimental results showed that when the ultrasonic power was 20 W, the bending strength of the CFRP prepared by the ultrasonic-assisted FFF manufacturing system could reach 111.5 MPa and the interlaminar shear strength could achieve 10.16 MPa, which increased by 33.11% and 21.5%, respectively, compared with the untreated specimens. However, the tensile strength of the specimen did not increase significantly compared with the bending strength and interlaminar shear strength. At the same power, the tensile strength of the specimen was only 166.8 MPa, which was only 5.6% higher than that of the untreated specimen. When the ultrasonic power exceeded 20 W, the mechanical properties of the CCFRPLA deteriorated. This may be because excessive ultrasonic power damaged the main structure of the fibers and weakened the interface performance between fibers and matrix, leading to a partial fracture of the fibers and affecting their overall structural performance. In addition, we explored the enhancement mechanism of ultrasonic vibration on the molecular scale. Ultrasonic vibration can cause the alternative fracture of PLA molecular chains and form PLA chains of different lengths. In the MD analysis, ultrasonic vibration enhanced the activity of PLA molecular chains and changed their conformation, which promoted entanglement, infiltration, and interaction among matrix molecular chains and between the matrix and fibers. This increased the density of the PLA matrix and reduced the interface distance between the CF and PLA matrix, thus improving the interface binding energy between the CF and PLA matrix and the separation resistance of matrix layers and ultimately enhancing the bending strength and interlaminar shear strength of the CCFRPLA. However, the improvement of interfacial properties between the CF and PLA matrix, as well as between interlaminar layers, has a relatively small promoting effect on the overall tensile strength of CCFRPLA. The MD simulation results agree with the experimental results, illustrating the effectiveness of ultrasonic vibration in improving the mechanical properties of CCFRPLA.

## Figures and Tables

**Figure 1 polymers-15-02554-f001:**
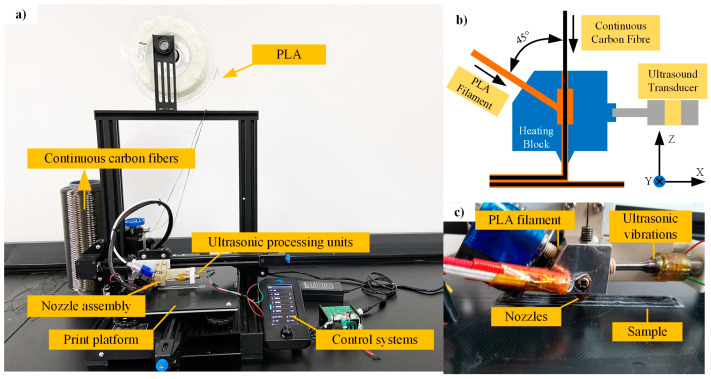
The ultrasonic online vibration-assisted CFRP AM platform. (**a**) The composition diagram of the designed platform. (**b**) The specific structure of the ultrasonic assisted nozzle. (**c**) The printing process of CFRP through the designed platform.

**Figure 2 polymers-15-02554-f002:**
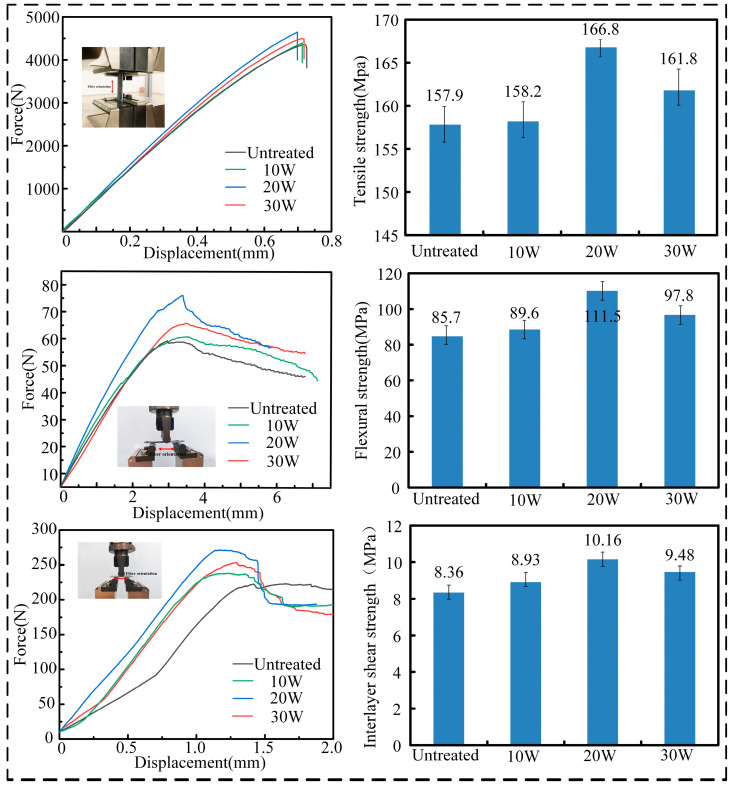
Mechanical tests and experimental results of CFRP specimens.

**Figure 3 polymers-15-02554-f003:**
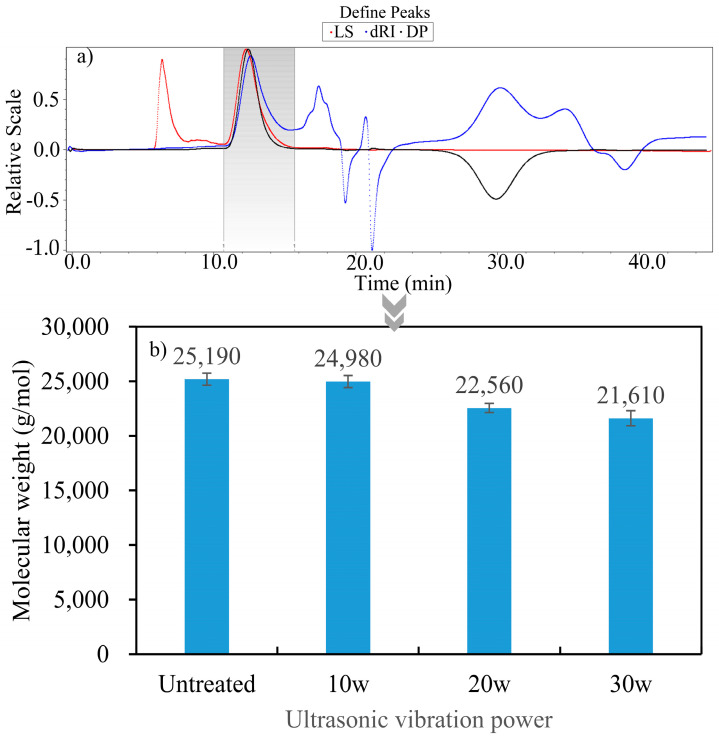
The weight of the PLA molecular chain treated with different ultrasonic vibration powers. (**a**) The GPC gel chromatogram of PLA treated with 10 W ultrasonic power. (**b**) The molecular weight of PLA treated with different ultrasonic vibration powers.

**Figure 4 polymers-15-02554-f004:**
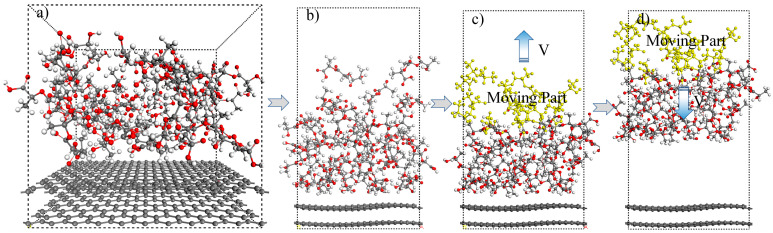
The MD simulation model and the ultrasonic vibration settings. (**a**) The original MD simulation model of PLA/CF. (**b**) The calculation model with a 20 Å vacuum layer on the PLA surface. (**c**) The upward moving of the defined moving part with an assigned initial velocity. (**d**) The downward moving of the defined moving part with the assigned initial velocity.

**Figure 5 polymers-15-02554-f005:**
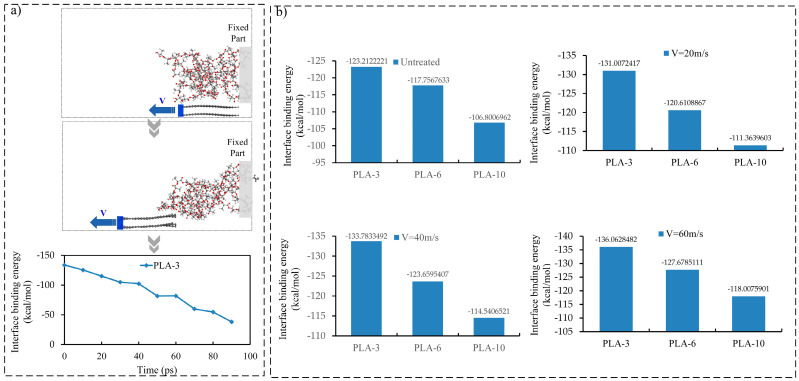
The detachment process of carbon fiber from the PLA matrix and the interface binding energy between the CF and PLA matrix. (**a**) The MD detachment process of CF from the PLA-3 matrix. (**b**) The interface binding energy between the CF and PLA matrix with different degrees of polymerization under different velocities.

**Figure 6 polymers-15-02554-f006:**
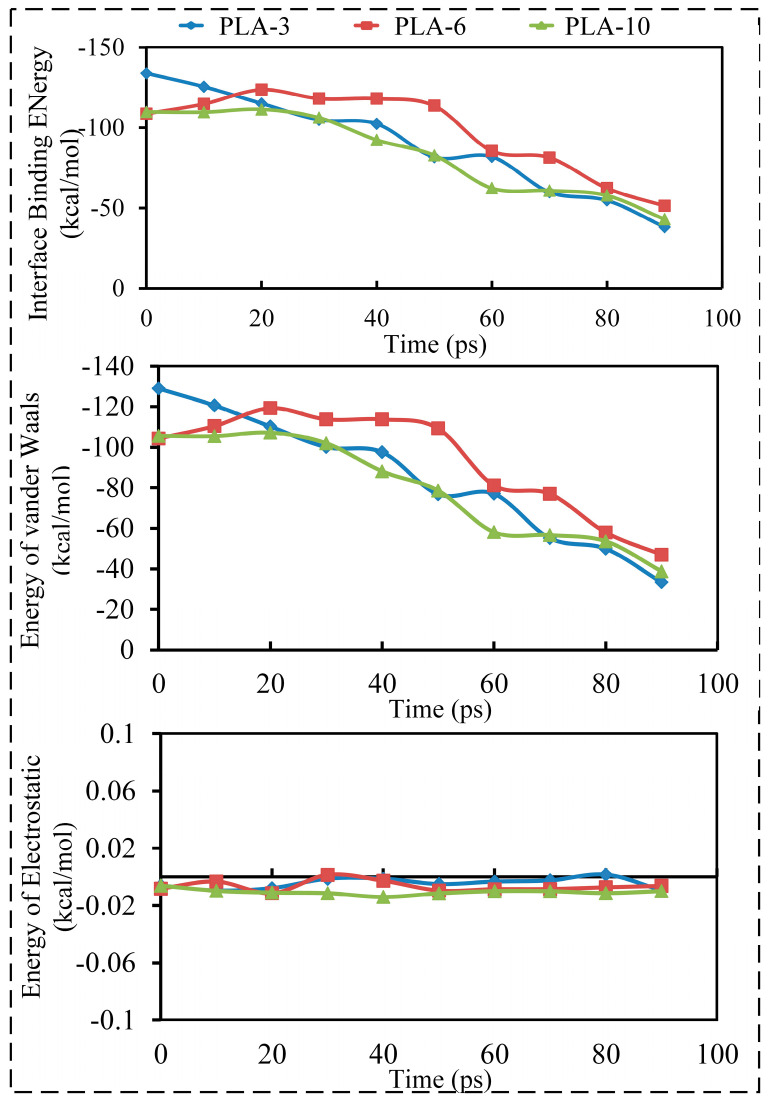
The changes of the interface binding energy and the components of the interface binding energy with time during the CF extraction process from the PLA matrix.

**Figure 7 polymers-15-02554-f007:**
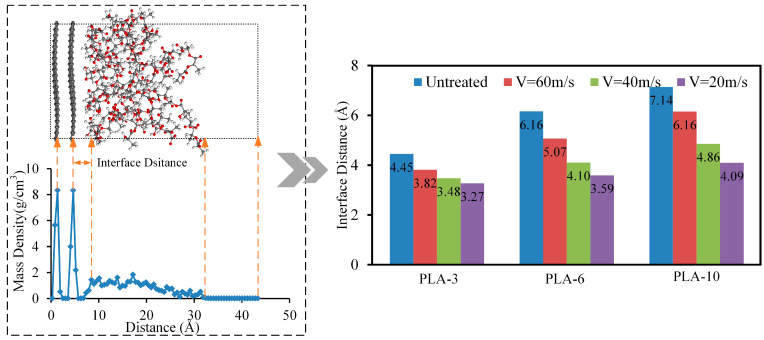
The interface distance between the CF and PLA under different ultrasonic velocities and matrix polymerization degrees.

**Figure 8 polymers-15-02554-f008:**
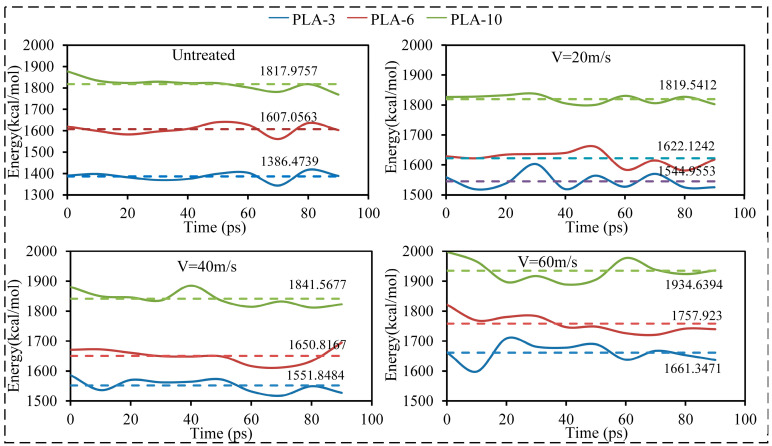
The changes in PLA matrix energy with different degrees of polymerization under different ultrasonic speed treatments during CF/PLA extraction.

**Figure 9 polymers-15-02554-f009:**
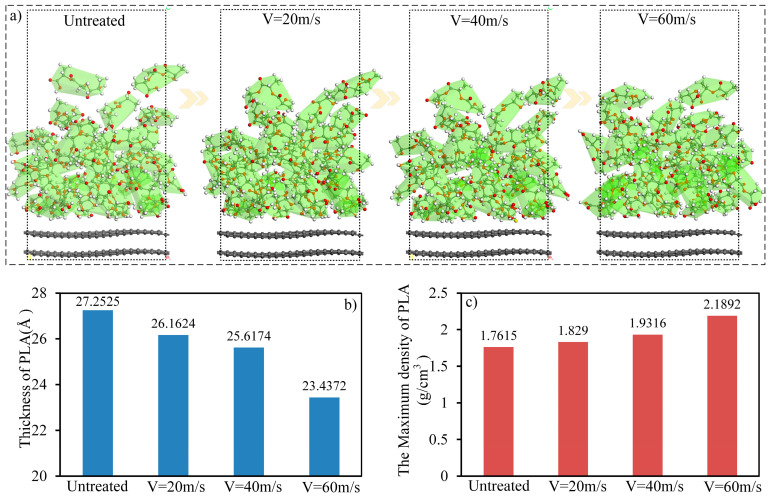
Changes in conformation, matrix thickness, and density of PLA with different ultrasonic speeds. (**a**) The PLA conformation changes with ultrasonic speed. (**b**) The matrix thickness changes with ultrasonic speed. (**c**) The maximum density changes with ultrasonic speed.

**Figure 10 polymers-15-02554-f010:**
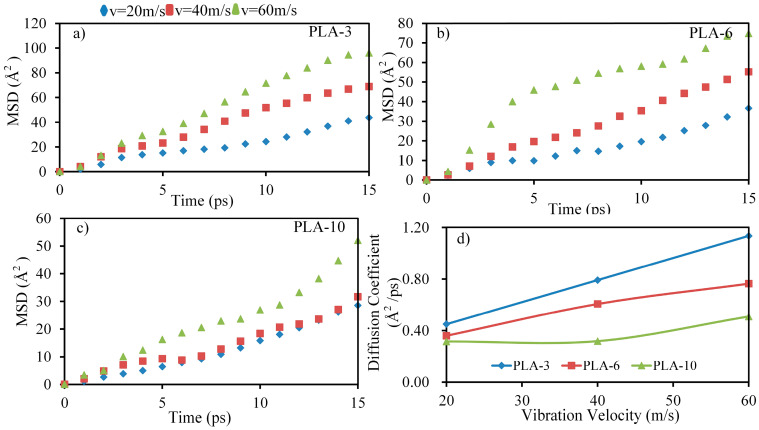
The MSD of PLA molecular chains with different ultrasonic velocities. (**a**) The MSD values of PLA-3 in the MD model assigned with different initial speeds. (**b**) The MSD values of PLA-6 in the MD model assigned with different initial speeds. (**c**) The MSD values of PLA-10 in the MD model assigned with different initial speeds. (**d**) The diffusion coefficient of PLA in the MD model assigned with different initial speeds.

**Figure 11 polymers-15-02554-f011:**
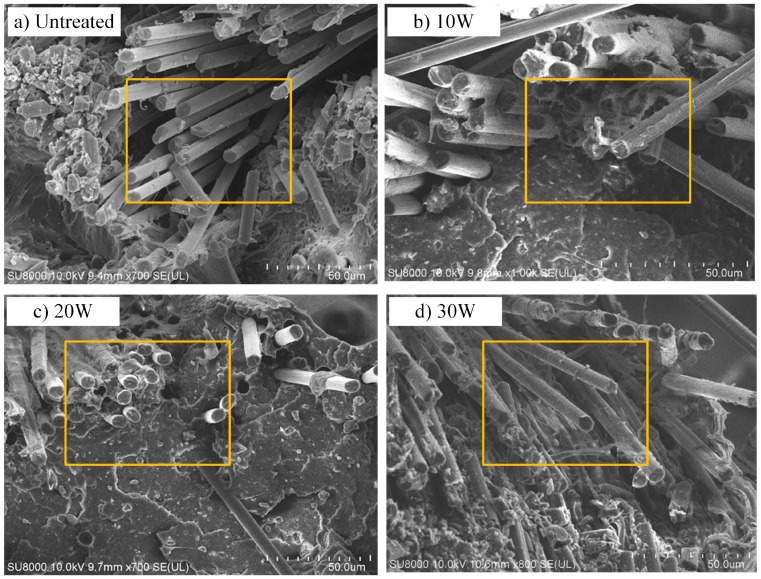
The cross-section SEM of the CCFRP fabricated with different ultrasonic powers.

**Table 1 polymers-15-02554-t001:** Process parameters of the specimen printing process.

Printing Settings	Values
Nozzle diameter	1.0 mm
Fan use (cooling)	75%
Extruder temperature	210 °C
Build platform temperature	80 °C
Printing speed	180 mm/min
First layer printing speed	144 mm/min
Fiber orientation	Unidirectional 0°
Infill ratio	100%
Layer height	1.0 mm

## Data Availability

The data presented in this study are available on request from the corresponding author.

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
