# Peer review of "Improving the Mechanical Properties of CCFRPLA by Enhancing the Interface Binding Energy and Strengthening the Anti-Separation Ability of a PLA Matrix"

_polymers, 2023, doi:10.3390/polym15112554_

Round 1
Reviewer 1 Report
The authors present a beautiful study which combines experiment and theory. I consider the manuscript as perfectly suited for publication in Polymers. However I have some minor points which should be answered:
- It seems to me, that the desired goal which is also expressed in the title, namely improving the mechanical properties of CFRP is not fully reached. The Conclusions could be formulated more cautiously in this respect.
- It is a bit unusual that the description of the method is given in the Discussion session (3.2. MD models and ultrasonic vibration settings).
- It is not clear what we learn from Figure 11 and how it is connected to the theoretical findings. The structures shown have a size of 50000 nm while the theoretical model systems have a size of roughly 5 nm.
- The literature list contains almost only Asian references. In this field, there is also a strong effort in Europe and the US which is completely neglected. (I admit that we often ignore Asian papers without cause.)
Author Response
Title: Improving mechanical properties of CCFRPLA by enhancing the interface binding energy and strengthening the anti-separation ability of PLA matrix
Reviewer 1:
Reply to the reviewers’ comments
The authors sincerely appreciate reviewers’ valuable comments. After thoroughly discussion and experiments, we have modified or added some part of the paper according to the suggestions in yellow.
The replies to these comments are written in red as follows.
It seems to me, that the desired goal which is also expressed in the title, namely improving the mechanical properties of CFRP is not fully reached. The Conclusions could be formulated more cautiously in this respect.
Thank you for your comments.
I have revised the conclusions according to the experimental results and added it to the abstract and conclusion parts, shown as following:
The bending strength and interlaminar shear strength of the specimen treated with 20 W ultrasonic vibration reached 111.5 MPa and 10.16 MPa, respectively, 33.11% and 21.5% higher than those of the untreated specimen, consistent with the molecular dynamics simulations and confirmed the effectiveness of ultrasonic vibration in improving the flexural and interlaminar properties of the CCFRPLA.
This increased the density of the PLA matrix and reduced the interface distance between the CF and PLA matrix, thus improving the interface binding energy between the CF and PLA matrix and the separation-resistance of matrix layers, and ultimately enhancing the bending strength and interlaminar shear strength of the CCFRPLA. However, the improvement of interfacial properties between CF and PLA matrix, as well as between interlaminar layers, has a relatively small promoting effect on the overall tensile strength of CCFRPLA.
It is a bit unusual that the description of the method is given in the Discussion session (3.2. MD models and ultrasonic vibration settings)
Thank you for your valuable advice.
I have revised the description of the MD simulation method in 3.2 and added in the paper.
According to the GPC results and considering the effect of ultrasonic vibration on the molecular chains of PLA matrix, three different PLA molecular chains with polymerization degrees of 3 (PLA-3), 6 (PLA-6), and 10 (PLA-10) were constructed in this paper and the corresponding molecular dynamics models with these three molecular chain lengths were constructed in the commercial software Materials Studio, as shown in Fig. 4a. In the MD models, the graphene was utilized as a substitute for CF.
A certain initial velocity was assigned to the moving part at the surface of the relaxed model in the Z-direction, as exhibited in the yellow part of Fig. 4c, representing the effect of ultrasonic vibration. Subsequently, a dynamic calculation of 50 ps with a time step of 1 fs was performed on the model in the NVT ensemble. After the calculation was completed, the last frame of the process was taken, and an initial velocity opposite to the previous direction was applied to the same PLA molecular chains to continue another 50 ps dynamic calculation under the same conditions, as shown in Fig. 4d. Changing the initial velocity assigned to the PLA molecular chains to explore the effect of ultrasonic powers on the performance of CCFRPLA, and three different initial velocities of 20, 40, and 60 m/s were set to conduct the above molecular dynamics research.
It is not clear what we learn from Figure 11 and how it is connected to the theoretical findings. The structures shown have a size of 50000 nm while the theoretical model systems have a size of roughly 5 nm.
Thank you very much.
SEM images are used to observe the bonding between PLA matrix and CF at a mesoscale, as well as the distribution of fibers in the matrix and verify the analysis results of molecular dynamics. The corresponding part has been added in this paper.
As Fig. 11a to Fig. 11c illustrated, with ultrasonic power increasing, there were more and more PLA matrix attaching to the fiber surface, indicating the improvement of bonding force between CF and PLA resin with the increase of ultrasonic power, which catered to the binding energy shown in Fig. 5. Furthermore, the infiltration between the CF bundle and PLA matrix and the uniformity of CF fibers dispersion in the matrix significantly improved as the ultrasonic power increased, demonstrating the effectiveness of ultrasonic vibration in enhancing the mechanical properties of CFRP, which coincided with the properties of CCFRPLA exhibited in Fig.2. However, when the ultrasonic power increased to 30W, the resin attached to the fiber surface and the wettability between CF and PLA matrix decreased at the side of 20W and there was a fracture inside the fiber bundle, which meant that excessive ultrasonic power could weaken the interface between fibers and matrix, as well as the integrity of CCFRPLA composites and this result shared the same coincidence with Fig. 2.
The literature list contains almost only Asian references. In this field, there is also a strong effort in Europe and the US which is completely neglected. (I admit that we often ignore Asian papers without cause.)
Thank you very much. Your suggestions have been considered, some related Europe and the US references have been added to the paper.
As one of the most promising bio-degradable polymers, polylactic acid (PLA) had been widely used in fused filament fabrication (FFF) process as feedstock or matrix to fabricate CFRP due to its excellent environmental friendliness and mechanical property [7-9]. Moreover, given the excellent physical properties of CCFRPLA, it was widely used in aerospace, automotive, wind energy, and sports fields [9-11].Andreu added a heated roller to the printer to increase interlayer adhesion [21]. In this process, the added heated roller could apply homogeneous pressure forces to increase filament surface contact, and the applied heat could enable longer diffusion and neck growth between adjacent filaments. Todoroki adopted a stepwise manufacturing approach to overcome the defect of no reinforcing fibers in the lay-up direction [22]. In this method, a composite part with a through hole and a reinforcing bar with continuous carbon fibers were printed separately. Subsequently, the reinforcing bar was inserted into the through hole, and the two parts were fused together via resistive heating.Ultrasonic vibration during FFF printing was a non-chemical/non-thermal process, and there were no adverse effects on the final product [11]. Chen developed a printer head that vibrated the nozzle vertically to improve the vertical tensile strength of FFF parts [25]. The vibration in the vertical direction reduced the viscosity of the material and brought downward inertial force, enhancing the diffusion of the polymer chains and strengthening the inter-layer bonding strength and mechanical properties of FFF parts. Tofangchi explored the effect of ultrasonic vibration on interlayer adhesion of acrylonitrile butadiene styrene (ABS) and found that ultrasonic vibrations could result in an increase of up to 10% in the interlayer adhesion. This increase was attributed to the increase in polymer reptation due to ultrasonic vibration-induced relaxation of the polymer chains from secondary interactions in the interface regions [26]. Maidin et al. [27] supplied the vibration onto the printing platform to identify the impact of ultrasound vibration on the ABS polymer, and the experimental results indicated that both the tensile strength and Young’s modulus improved by 11.3% and 16.7% compared to the untreated specimen [28,29].
[7]R. A. Ilyas, M. Y. M. Zuhri, H. A. Aisyah, et al. Natural Fiber-Reinforced Polylactic Acid, Polylactic Acid Blends and Their Composites for Advanced Applications. Polymers, 2022, 14(1), 202.
[8]Víctor A. Ramírez-Elías, Noemi Damian-Escoto, Kyosung Choo, et al. Structural Analysis of Carbon Fiber 3D-Printed Ribs for Small Wind Turbine Blades. Polymers, 2022, 14(22), 4925.
[9]Zia Ullah Arif, Muhammad Yasir Khalid, Reza Noroozi, et al. Recent advances in 3D-printed polylactide and polycaprolactone-based biomaterials for tissue engineering applications. International Journal of Biological Macromolecules, 2022, 218, 930-968.
[11]Sachini Wickramasinghe, Truong Do and Phuong Tran. FDM-Based 3D Printing of Polymer and Associated Composite: A Review on Mechanical Properties, Defects and Treatments. 2020, 1529, 12071529.
[21]Alberto Andreu, Sanglae Kim , Jörg Dittus, et al. Hybrid material extrusion 3D printing to strengthen interlayer adhesion through hot rolling. Additive Manufacturing, 2022, 55, 102773.
[22]Akira Todoroki, Tatsuki Oasada, Masahito Ueda, et al. Reinforcing in the lay-up direction with self-heating for carbon fiber composites fabricated using a fused filament fabrication 3D printer. Composite Structures, 2021, 266, 113815.
[25]Fuhui Chen, Qian Xu, Fuwen Huang, et al. Effect of nozzle vibration at different frequencies on surface structures and tensile properties of PLA parts printed by FDM. Materials Letters, 2022, 325, 132612.
[26]Alireza Tofangchi, Pu Han , Julio Izquierdo, et al. Effect of Ultrasonic Vibration on Interlayer Adhesion in Fused Filament Fabrication 3D Printed ABS. Polymers, 2019, 315, 11020315.
[27]Maidin, S.; Muhamad, M.K.; Pei, E. Feasibility study of ultrasonic frequency application on fdm to improve parts surface finish. J. Teknol. 2015, 77, 27–35.

Reviewer 2 Report
The PLA matrix should be mentioned in the abstract and the title.
A novelty is not shown.
Significance is not shown.
A literature review on ultrasonic vibrations in CFRP composite manufacturing and properties needs to be included. The authors cite only one article and this is their article.
As PLA was chosen as the matrix, the Introduction should be extended with a literature review of PLA/CFRP composites - properties and applications.
From the PLA matrix perspective, the first sentence, "Continuous fiber-reinforced polymers (CFRP) have been widely used in aerospace, rail transit, and marine equipment because of their excellent features, ..." is confusing. PLA matrix composites are not suitable for such applications.
CFRP was obtained in an extruder, which dictates the specific type of products. It should be addressed in the Introduction.
Using the term cross-linking for PLA is unacceptable. PLA is a thermoplastic, not duroplastic.
How does ultrasonic vibration affect the structure of PLA? I think it should be talked about.
How does ultrasonic affect the processing temperature of PLA composites? I think it should be talked about.
PLA supplier is missing.
Author Response
Title: Improving mechanical properties of CCFRPLA by enhancing the interface binding energy and strengthening the anti-separation ability of PLA matrix
reviewer 2:
Reply to the reviewers’ comments
The authors sincerely appreciate reviewers’ valuable comments. After thoroughly discussion and experiments, we have modified or added some part of the paper according to the suggestions in yellow.
The replies to these comments are written in red as follows.
- The PLA matrix should be mentioned in the abstract and the title.
Thank you so much. The PLA matrix has been mentioned in the abstract and the title.
Title: Improving mechanical properties of CCFRPLA by enhancing the interface binding energy and strengthening the anti-separation ability of PLA matrix
Abstract: Additive manufacturing (AM) can produce almost any product shape through layered stacking. The usability of continuous fiber-reinforced polymers (CFRP) fabricated by AM, however, is restricted owing to the limitations of no reinforcing fibers in the lay-up direction and weak interface bonding between the fibers and matrix. This study presents molecular dynamics in conjunction with experiments to explore how ultrasonic vibration enhances the performance of continuous carbon fiber-reinforced polylactic acid (CCFRPLA). Ultrasonic vibration improves the mobility of PLA matrix molecular chains by causing alternative fractures of chains, promoting crosslinking infiltration among polymer chains, and interactions between carbon fibers and the matrix. The increase in entanglement density and conformational changes enhanced the density of the PLA matrix and strengthened its anti-separation ability. In addition, ultrasonic vibration shortens the distance between the molecules of the fiber and matrix, improving the van der Waals force and thus promoting the interface binding energy between them, which ultimately achieves an overall improvement in the performance of CCFRPLA. The bending strength and interlaminar shear strength of the specimen treated with 20 W ultrasonic vibration reached 111.5 MPa and 10.16 MPa, respectively, 33.11% and 21.5% higher than those of the untreated specimen, consistent with the molecular dynamics simulations and confirmed the effectiveness of ultrasonic vibration in improving the flexural and interlaminar properties of the CCFRPLA.
2.A novelty is not shown. Significance is not shown.
Thank you so much. The innovation of this article is to design an ultrasonic vibration online-assisted AM platform and explore the effect of ultrasonic vibration on the performance of CCFRPLA at the molecular scale. According to the molecular dynamics results, ultrasonic vibration improves the mobility of PLA matrix molecular chains by causing alternative fractures of chains, promoting crosslinking infiltration among polymer chains, and interactions between carbon fibers and the matrix. The increase in entanglement density and conformational changes enhanced the density of the PLA matrix and strengthened its anti-separation ability. In addition, ultrasonic vibration shortens the distance between the molecules of the fiber and matrix, improving the van der Waals force and thus promoting the interface binding energy between them, which ultimately achieves an overall improvement in the performance of CCFRPLA. The significance of this study is that through the designed ultrasonic vibration online-assisted AM platform, the bending strength and interlaminar shear strength of the CCFRPLA composites treated with 20 W ultrasonic vibration reached 111.5 MPa and 10.16 MPa, respectively, 33.11% and 21.5% higher than those of the untreated specimen, and the influence of ultrasonic power on CCFRPLA performance has been explored, providing a methodological basis for the selection of ultrasonic power and the improvement of CFRP performance.
- A literature review on ultrasonic vibrations in CFRP composite manufacturing and properties needs to be included. The authors cite only one article and this is their article.
Thank you very much. Your suggestions have been considered and the literature review on ultrasonic vibrations in CFRP composite manufacturing and properties have been added to the paper.
Ultrasonic vibration during FFF printing was a non-chemical/non-thermal process, and there were no adverse effects on the final product [11]. Chen developed a printer head that vibrated the nozzle vertically to improve the vertical tensile strength of FFF parts [25]. The vibration in the vertical direction reduced the viscosity of the material and brought downward inertial force, enhancing the diffusion of the polymer chains and strengthening the inter-layer bonding strength and mechanical properties of FFF parts. Tofangchi explored the effect of ultrasonic vibration on interlayer adhesion of acrylonitrile butadiene styrene (ABS) and found that ultrasonic vibrations could result in an increase of up to 10% in the interlayer adhesion. This increase was attributed to the increase in polymer reptation due to ultrasonic vibration-induced relaxation of the polymer chains from secondary interactions in the interface regions [26]. Maidin et al. [27] supplied the vibration onto the printing platform to identify the impact of ultrasound vibration on the ABS polymer, and the experimental results indicated that both the tensile strength and Young’s modulus improved by 11.3% and 16.7% compared to the untreated specimen [28,29].
- As PLA was chosen as the matrix, the Introduction should be extended with a literature review of PLA/CFRP composites - properties and applications.
Thank you so much. A literature review of PLA/CFRP composites - properties and applications have been added in the paper.
As one of the most promising bio-degradable polymers, polylactic acid (PLA) had been widely used in fused filament fabrication (FFF) process as feedstock or matrix to fabricate CFRP due to its excellent environmental friendliness and mechanical property [7-9], and the PLA/CFRP composites could be printed by a dual extrusion or co-extrusion method based on AM [10, 11]. Moreover, given the excellent physical properties of CCFRPLA, it was widely used in aerospace, automotive, wind energy, and sports fields [9-11].
- From the PLA matrix perspective, the first sentence, "Continuous fiber-reinforced polymers (CFRP) have been widely used in aerospace, rail transit, and marine equipment because of their excellent features, ..." is confusing. PLA matrix composites are not suitable for such applications.
Thank you so much. The first sentence, "Continuous fiber-reinforced polymers (CFRP) have been widely used in aerospace, rail transit, and marine equipment because of their excellent features.”, is utilized to reflect the excellent performance of CFRP and its importance in industry and daily life. According to my studies and a lot of reading literature, PLA/CFRP composites have been widely used in aerospace, automotive, wind energy, and sports fields.
[3] Ziang Jin, Zhenyu Han, Cheng Chang, et al. Review of methods for enhancing interlaminar mechanical properties of fiber-reinforced thermoplastic composites: Interfacial modification, nano-filling and forming technology. Composites Science and Technology, 2022, 228, 109660.
[7]R. A. Ilyas, M. Y. M. Zuhri, H. A. Aisyah, et al. Natural Fiber-Reinforced Polylactic Acid, Polylactic Acid Blends and Their Composites for Advanced Applications. Polymers, 2022, 14(1), 202.
[8]Víctor A. Ramírez-Elías, Noemi Damian-Escoto, Kyosung Choo, et al. Structural Analysis of Carbon Fiber 3D-Printed Ribs for Small Wind Turbine Blades. Polymers, 2022, 14(22), 4925.
[9]Zia Ullah Arif, Muhammad Yasir Khalid, Reza Noroozi, et al. Recent advances in 3D-printed polylactide and polycaprolactone-based biomaterials for tissue engineering applications. International Journal of Biological Macromolecules, 2022, 218, 930-968.
[10]Fuji Wang, Zhongbiao Zhang, Fuda Ning, et al. A mechanistic model for tensile property of continuous carbon fiber reinforced plastic composites built by fused filament fabrication. Additive Manufacturing, 2020, 32, 101102.
[11]Sachini Wickramasinghe, Truong Do and Phuong Tran. FDM-Based 3D Printing of Polymer and Associated Composite: A Review on Mechanical Properties, Defects and Treatments. 2020, 1529, 12071529.
- CFRP was obtained in an extruder, which dictates the specific type of products. It should be addressed in the Introduction.
Thank you so much. Your suggestions have been considered and the specific type of products have been added in the paper.
There was a through-hole in the center of the heating block through which the continuous fibers passed during printing. A slanted hole on the side at a 45° from the center hole was designed, and the polymer matrix entered through this hole and melted in the mixing chamber during printing, as shown in Fig. 1b. In the co-extrusion process of CCFRPLA, the PLA resin filament and the continuous carbon fibers were separately supplied to the printer head. The PLA filament got molten inside the heated nozzle, and when the continuous carbon fibers was passed through the nozzle, it got impregnated by the PLA resin and extruded from the bottom nozzle, as shown in Fig. 1c.
Tensile and interlaminar shear tests were carried out in accordance with ISO 527:1997 and ISO 14130:1997, respectively, where the CCFRPLA specimen dimensions for the tensile test were 250 mm × 14 mm × 2 mm, and the fiber direction in the specimen was the same as the tensile direction. The specimen dimensions for the interlaminar shear test were 20 mm × 10 mm × 2 mm, and the direction of the carbon fiber in the CFRP was consistent with the long side of the specimen. In addition, the bending strength of specimens under different ultrasonic powers was studied, and the specimen for the three-point bending test was of a non-standard design with the dimensions 60 mm × 10 mm × 2 mm.
- Using the term cross-linking for PLA is unacceptable. PLA is a thermoplastic, not duroplastic.
Thank you so much. Your suggestions have been considered and the term cross-linking has been replaced with entanglement.
- How does ultrasonic vibration affect the structure of PLA? I think it should be talked about.
Thank you so much. Your suggestions have been considered and the effect of ultrasonic vibration on PLA structure has been added in the paper.
As shown in Fig. 9a, with an increase in the ultrasonic vibration amplitude, the conformation of the PLA molecular chains changes and PLA molecular chains become more stretched, which promotes entanglement among molecular chains, reducing the porosity inside the matrix, and increasing the density of the matrix.
- How does ultrasonic affect the processing temperature of PLA composites? I think it should be talked about.
Thank you so much. In the printing process of CCFRPLA composite, due to the thermocouples and the temperature setting, the temperature of PLA in the nozzle is a certain value and is not affected by ultrasonic vibration.
- PLA supplier is missing.
Thank you so much. The PLA supplier message has been added in the paper.
The PLA (Shenzhen Creativity 3D Technology Co. Ltd., China) was chosen as the resin matrix material in this study owing to its excellent mechanical properties and environmental friendliness.

Round 2
Reviewer 2 Report
Accept in present form.